# Cross-Entropy Method for Content Placement and User Association in Cache-Enabled Coordinated Ultra-Dense Networks

**DOI:** 10.3390/e21060576

**Published:** 2019-06-08

**Authors:** Jia Yu, Ye Wang, Shushi Gu, Qinyu Zhang, Siyun Chen, Yalin Zhang

**Affiliations:** 1Communication Engineering Research Centre, Harbin Institute of Technology (Shenzhen), HIT Campus of University Town of Shenzhen, Shenzhen 518055, China; 2Peng Cheng Laboratory, Shenzhen 518055, China; 3School of Electronic and Communication Engineering, Shenzhen Polytechnic, Shenzhen 518055, China

**Keywords:** ultra dense network, cross-entropy, proactive caching, user association, CoMP

## Abstract

Due to the high splitting-gain of dense small cells, Ultra-Dense Network (UDN) is regarded as a promising networking technology to achieve high data rate and low latency in 5G mobile communications. In UDNs, each User Equipment (UE) may receive signals from multiple Base Stations (BSs), which impose severe interference in the networks and in turn motivates the possibility of using Coordinated Multi-Point (CoMP) transmissions to further enhance network capacity. In CoMP-based Ultra-Dense Networks, a great challenge is to tradeoff between the gain of network throughput and the worsening backhaul latency. Caching popular files on BSs has been identified as a promising method to reduce the backhaul traffic load. In this paper, we investigated content placement strategies and user association algorithms for the proactive caching ultra dense networks. The problem has been formulated to maximize network throughput of cell edge UEs under the consideration of backhaul load, which is a constrained non-convex combinatorial optimization problem. To decrease the complexity, the problem is decomposed into two suboptimal problems. We first solved the content placement algorithm based on the cross-entropy (CE) method to minimize the backhaul load of the network. Then, a user association algorithm based on the CE method was employed to pursue larger network throughput of cell edge UEs. Simulation were conducted to validate the performance of the proposed cross-entropy based schemes in terms of network throughput and backhaul load. The simulation results show that the proposed cross-entropy based content placement scheme significantly outperform the conventional random and Most Popular Content placement schemes, with with 50% and 20% backhaul load decrease respectively. Furthermore, the proposed cross-entropy based user association scheme can achieve 30% and 23% throughput gain, compared with the conventional *N*-best, No-CoMP, and Threshold based user association schemes.

## 1. Introduction

Inspired by the development of intelligent terminal such as smart phones, the demand for data traffic in mobile communication systems is exponentially growing. To cater for this demand, a 1000-fold improvement of capacity per area in the next generation of mobile communication system (5G) compared to 4G is required. An Ultra-Dense Network (UDN) is capable of significantly improving the capacity per area under the limited spectrum resource due to the high splitting-gain of densely located small cells and is widely considered as one of the most promising techniques in the coming 5G. It also benefits load balance between Base Stations (BSs) since Small Base Stations (SBSs) can offload data traffic of Macro Base Stations (MBSs). Nevertheless, due to the short distance between BSs, the intercell interference in UDNs is severe, therefore making the user experience unsatisfactory. Coordinated Multi-Point (CoMP) transmissions technique is widely studied in academia and the industry, which can leverage the cooperation of multiple BSs to enhance the signal to interference and noise ratio (SINR), to counteract intercell interference and to enhance network capacity in UDNs.

Despite remarkable performance gain in network capacity, congestion on backhaul links caused by CoMP risks the mobile communication systems. In order to cooperatively serve users, BSs need to fetch more files from the Core Network (CN) via backhaul links in a CoMP-employed system, which brings heavy load to backhaul links between BSs and the CN and probably results in congestion. One way to alleviate backhaul load is to cache popular files on BSs. When BSs cache files requested by users, it does not need to fetch files from the CN, so the backhaul load can be dramatically reduced. In [1], an architecture based on distributed caching of content in SBSs was presented. Works on proactive caching in UDNs concentrate on two major issues: Content placement and content distribution.

Content placement focuses on how to distribute popular and hotspot files to the BSs’ caching unit whose capacity is limited. In [2], an optimal content placement strategy is proposed to maximize the hit rate. In [3], the problem of content placement is studied to maximize energy efficiency. In [4], an approximation algorithm is proposed to jointly optimize routing and caching policy to maximize the fraction of requested files cached locally. In [5], a distributed algorithm is proposed to investigate content placement and user association jointly. In [6], a content placement strategy is investigated based on reinforcement learning. In [7], a content placement strategy is proposed under cooperation schemes of maximum ratio transmission (MRT) and zero-forcing beamforming (ZFBF). In [8], a caching space allocation scheme is proposed to improve the hit rate based on the categories of contents and UEs.

Content distribution is the study of how to associate UEs and BSs to improve the hit rate. In [9], the user association problem is modeled as an one-to-many game problem, based on which algorithm is proposed to maximize the average download rate under a given content placement strategy. In [10], an user association algorithm under a given content distribution in a CoMP enabled network is proposed to minimize the backhaul load under a guaranteed rate requirements of UEs. In [11], the content caching and user association schemes are proposed on two different scales: The caching algorithm operates in a long time scale and the user association algorithm operates frequently. In [12], user association is investigated to tradeoff between load balancing and backhaul savings in UDN.

To the best of our knowledge, related works on the content placement caching and user association have been investigated separately in small-scale networks. We are thus motivated to jointly investigate the tasks of caching and user association in more realistic large-scale UDNs. The main contributions of this paper are summarized as follows.
The problem of content placement and user association is investigated jointly in large-scale cache-enabled coordinated ultra dense networks. We formulate the problem as a constrained non-convex combinatorial programming problem to maximize network throughput of cell edge UEs under the consideration of the backhaul load;A two-step heuristic algorithm based on the cross-entropy (CE) method is proposed to solve the problem: A content placement strategy is first proposed based on cross entropy under the assumption of the conventional *N*-Best scheme; given the proposed content placement strategy, a user association algorithm is then proposed based on the cross-entropy method. Extensive simulations are conducted to evaluate the performance of the proposed approach. Simulations are conducted to validate the performance of the proposed cross-entropy based schemes in terms of network throughput and backhaul load. Simulation results show that the proposed caching and user association algorithms can reduce backhaul load and improve network throughput of cell edge UEs simultaneously.

The rest of this paper is organized as follows. The system model is established in Section 2 with some basic assumptions. In Section 3, we formulate the problem and propose the algorithms. In Section 4, simulation results are presented and the system performance is evaluated. Section 5 concludes this paper.

## 2. System Model

### 2.1. Network

In this paper, we consider a heterogeneous Ultra-Dense Network consisting of $N_{\mathrm{MBS}}$ Macro BSs (MBS) and $N_{\mathrm{SBS}}$ Small Base Stations (SBS). The Macro BSs are uniformly distributed to provide coverage and to support capacity. The small BSs are randomly distributed within the covering area, following a Poisson Point Process (PPP) with a density of $\lambda_{\mathrm{SBS}}$. Let $B^{\Omega }=\{b|b=1,2,\cdots ,|B^{\Omega }\left|\right\}$ denote the set of BSs consisting of both Macro BSs and Small BSs, where $|B^{\Omega }|=N_{\mathrm{MBS}}+N_{\mathrm{SBS}}$ is the total number of BSs.

UEs are randomly distributed following a PPP with density of $\lambda_{U}$. The set of UEs is denoted by $M^{\Omega }=\{m|m=1,2,\cdots ,|M^{\Omega }\left|\right\}$. UEs located at the edges of cells usually suffer from severe intercell interference and low capacity. To reduce the interference and enhance peak data rates of cell edge users, joint transmission Coordinated Multi-Point (termed as JT CoMP) is considered in the network architecture, which allows multiple BSs in the neighborhood to cooperatively serve a specific UE simultaneously.

The association relationship between UEs, *m*, and BS, *b*, is denoted by a bit number $x_{m,b}\left(m\in M^{\Omega },b\in B^{\Omega }\right)$ defined as:(1)$$x_{m,b}=\left\{\begin{array}{cc} 1 & \mathrm{UE}\;m\;\mathrm{is}\;\mathrm{associated}\;\mathrm{with}\;\mathrm{BS}\;b\\ 0 & \mathrm{otherwise} \end{array}\right..$$

Thus, the entire association result between BSs and UEs in the considering network can be presented by:(2)$$X=\left[\begin{array}{cccc} x_{1,1} & x_{1,2} & \cdots & x_{1,\left|B^{\Omega }\right|}\\ x_{2,1} & x_{2,2} & \cdots & x_{2,\left|B^{\Omega }\right|}\\ \vdots & \vdots & \ddots & \vdots \\ x_{\left|M^{\Omega }\right|,1} & x_{\left|M^{\Omega }\right|,2} & \cdots & x_{\left|M^{\Omega }\right|,\left|B^{\Omega }\right|} \end{array}\right].$$

The SINR of a specific UE in a downlink transmission can be presented in terms of $x_{m,b}$ as follows,
(3)$$\Gamma_{m}=\frac{\sum_{b\in B^{\Omega }}x_{m,b}P_{b}g_{m,b}}{\sum_{b^{\prime }\in B^{\Omega }}\left(1-x_{m,b^{\prime }}\right)P_{b^{\prime }}g_{m,b}+\sigma^{2}},$$
where $P_{b}$ is the transmit power of BS *b*, $g_{m,b}$ is the channel gain between BS *b* and UE *m*, and $\sigma^{2}$ is the variance of additive white Gaussian noise (AWGN). Equation (Equation 3) suggests that the more BSs are associated with UE *m*, the better service it can obtain. However, the necessary overhead to accomplish a JT CoMP transmission involving too much BSs is unacceptable. Thus, it is better to narrow the associating BSs of a UE into *N* BSs in close proximity.

As in 4G LTE and 5G NR (New Radio) wireless standards, resource block (RB) is considered the unit of time and spectrum resource for allocation in this paper. Assume that the bandwidth of each RB is *W* and the total number of RBs is $N_{\mathrm{RB}}$. Each UE in the network can occupy a part of resource for transmission. Let $\beta_{m}$ denote the proportion that the resource assigned to UE *m* out of all. Then the data rate of a downlink transmission to UE *m* can be given by:(4)$$R_{m}=W\left\lfloor \beta_{m}N_{\mathrm{RB}}\right\rfloor \log_{2}\left(1+\Gamma_{m}\right),$$
where ⌊x⌋ is the minimum integer smaller than or equal to *x*; $\Gamma_{m}$ is defined by Equation (Equation 3).

Let $B_{m}^{\Omega }\subseteq B^{\Omega }$ denote the set consisting of BSs associated with UE *m*, i.e., $B_{m}^{\Omega }=\left\{b|b\in B^{\Omega },\;\mathrm{and}\;x_{m,b}=1\right\}$. Similarly, let $M_{b}^{\Omega }\subseteq M^{\Omega }$ denote the set consisting of UEs associated with BS *b*,  i.e., $M_{b}^{\Omega }=\left\{m|m\in M^{\Omega },\;\mathrm{and}\;x_{m,b}=1\right\}$. The resource that BS *b* can assign to UE *m* should be no more than $\frac{1}{|M_{b}^{\Omega }|}$, where $|M_{b}^{\Omega }|$ represents the total number of UEs associated with BS *b*. In the case where JT CoMP is employed, the resource that UE *m* can be obtained is restricted by the most heavy-loading BS among those associated with UE *m*. As a result, the proportion $\beta_{m}$ can be given by:(5)$$\beta_{m}=\min\left\{\frac{1}{\left|M_{b}^{\Omega }\right|},b\in B_{m}^{\Omega }\right\}.$$

### 2.2. Caching

UDNs can benefit from caching popular files on BSs in terms of throughput, delay, and traffic load on backhaul links. It is obvious that the more files cached on BSs, the better performance a network can achieve. The cache-enabled heterogeneous UDN we considered in this paper is illustrated as Figure 1. For simplicity, we assume that the files requested by all UEs in the networks are restricted into the set of $F^{\Omega }=\{f|f=1,2,\cdots ,|F^{\Omega }\left|\right\}$, and each file is in the same size of $F_{\max}$ bits. We assume the popularity of files follows Zipf distribution [13]. Let $p_{f}$ denote the probability mass function of popularity random variable *F*. The probability that file *f* is requested can be given by:(6)$$\begin{array}{c} p_{f}=\frac{{\left(\sum_{f=1}^{|F^{\Omega }|}f^{-\gamma }\right)}^{-1}}{f^{\gamma }}, \end{array}$$
where γ is the Shape Factor (SF) indicating the correlation between requests of UEs [13]. It is seen that the larger the shape factor γ is, the smaller the probability mass function $p_{f}$ would be, the lower probability file *f* is requested out of the caching set $F^{\Omega }$.

Define a caching index vector $y_{b}=\left[y_{b,1},y_{b,2},\cdots ,y_{b,f},\cdots y_{b,|F^{\Omega }|}\right]$, where $y_{b,f}$ indicates whether BS *b* caches file *f* in the caching set $F^{\Omega }$ or not. More specifically,
(7)$$y_{b,f}=\left\{\begin{array}{cc} 1 & \mathrm{file}\;f\;\mathrm{is}\;\mathrm{cached}\;\mathrm{on}\;\mathrm{BS}\;b\\ 0 & \mathrm{otherwise} \end{array}\right..$$

Then the File-BS caching matrix can be denoted by:(8)$$Y=\left[\begin{array}{cccc} y_{1,1} & y_{1,2} & \cdots & y_{\left|B^{\Omega }\right|,\left|F^{\Omega }\right|}\\ y_{2,1} & y_{2,2} & \cdots & y_{\left|B^{\Omega }\right|,\left|F^{\Omega }\right|}\\ \vdots & \vdots & \ddots & \vdots \\ y_{\left|B^{\Omega }\right|,1} & y_{\left|B^{\Omega }\right|,2} & \cdots & y_{\left|B^{\Omega }\right|,\left|F^{\Omega }\right|} \end{array}\right].$$

As for UEs, we define a row vector  $q_{m}=\left[q_{m,1},q_{m,2},\cdots ,q_{m,f},\cdots \right]$, where $q_{m,f}$ represents the request of UE *m* to file *f*, i.e.,
(9)$$q_{m,f}=\left\{\begin{array}{cc} 1 & \mathrm{file}\;f\;\mathrm{is}\;\mathrm{requested}\;\mathrm{by}\;\mathrm{UE}\;m\\ 0 & \mathrm{otherwise} \end{array}.\right.$$

Suppose a UE can request one and only one file at each time, then we have $\sum_{f}q_{m,f}=1$, $\forall m\in M^{\Omega }$.

Then  $q_{m}y_{b}^{T}$ represents whether the file requested by UE *m* is caching on BS *b*, where $v^{T}$ represents the transpose of the vector v.
(10)$$q_{m}y_{b}^{T}=\left\{\begin{array}{cc} 1 & \mathrm{file}\;f\;\mathrm{requested}\;\mathrm{by}\;\mathrm{UE}\;m\;\mathrm{is}\;\mathrm{on}\;\mathrm{BS}\;b\\ 0 & \mathrm{otherwise} \end{array}\right..$$

If UE *m* is associated with BS *b* (i.e., $x_{m,b}=1$), we say that BS *b* misses a file if file *f* requested by UE *m* is not cached in it.

If BSs miss a file, they need to fetch the file from the Core Network (CN). We assume a centralized deployment of the considered UDN, where each BS is directly connected to the CN via backhaul links. The backhaul load is defined as the traffic carried by backhaul links between BSs and the CN [14]. In the case that BS *b* misses a file requested by UE *m*, BS *b* have to fetch the file from the CN through the backhaul link, which aggravates the backhaul load of BS *b* inevitably. In reality, the capacity of backhaul links is usually limited. Congestion occurs when the backhaul load of BS *b* exceeds the backhaul capacity $C_{b}^{\max}$.

Let $U_{\mathrm{back}}$ represent the increase of backhaul load due to fetching a file from the CN. The backhaul load caused by UE *m* can be given by:(11)$$V_{m}=\sum_{b\in B_{m}^{\Omega }}\left(1-q_{m}y_{b}^{T}\right)U_{\mathrm{back}}.$$

It is obvious that the more files BSs cache (i.e., the less files BSs miss), the less heavier the backhaul load will be.

### 2.3. Delay

In addition to reducing backhaul load, caching also benefits from reducing the time delay of transmissions. In this paper, we consider the average time delay of UEs, which consists of two major parts: Wireless propagation delay and backhaul delay. Let $d_{1}$ denote the average wireless propagation delay of a UDN which is related to the size of a file $F_{\max}$ and the data rate of transmission.
(12)$$d_{1}=\frac{1}{\left|M^{\Omega }\right|}\sum_{m\in M^{\Omega }}\frac{F_{\max}}{R_{m}}.$$

Backhaul delay, denoted by $d_{2}$, is related to whether the files are hit by the associating BSs of UEs. Due to joint transmission of CoMP, the backhaul delay for a specific UE *m* occurs when the requested file is not cached on all its associated BSs. That is, the minimum operation should be applied on backhaul delays due to file transmission between the associated BSs and the core network. More specifically, $d_{2}$ can be represented as follows,
(13)$$d_{2}=\frac{1}{\left|M^{\Omega }\right|}\sum_{m\in M^{\Omega }}\left(\min\left\{\frac{F_{\max}}{U_{\mathrm{back}}},\sum_{b\in B_{m}^{\Omega }}\left(1-q_{m}y_{b}^{T}\right)\frac{F_{\max}}{U_{\mathrm{back}}}\right\}\right).$$

As a result, the total time delay of the network is:(14)$$D=d_{1}+d_{2}.$$

## 3. Problem Formulation

### 3.1. Mathematical Formulation

In this paper, we aim to develop the optimal solution of content placement and user association that balances network throughput and the backhaul load simultaneously. Considering the fairness of UEs, the network throughput of cell edge UEs is used as the performance metric. Thus the objective function is a tradeoff between network throughput of cell edge UEs and backhaul load.
(15)$$\begin{array}{c} \max_{X,Y}\;\;\sum_{m\in M^{\Omega }}\log_{10}\left(R_{m}\right)-\lambda \frac{\sum_{m\in M^{\Omega }}V_{m}}{\left|M^{\Omega }\right|} \end{array}$$
$$\begin{array}{c} \begin{array}{ccc} s.t. & C1: & 1\leq |B_{m}^{\Omega }|\leq N,\forall m\in M^{\Omega }\\ & C2: & \sum_{m}x_{m,b}\leq N_{\mathrm{RB}},\forall b\in B^{\Omega },\forall m\in M^{\Omega }\\ & C3: & \sum_{m}x_{m,b}\left(1-q_{m}y_{b}^{T}\right)U_{\mathrm{back}}\leq C_{b}^{\max},\forall b\in B^{\Omega } \end{array} \end{array}$$
where the constraint C1 indicates that a specific UE *m* should be served by at least one BS, meanwhile the total number of BSs that cooperatively serve a specific UE should not be more than a given number *N* by considering the tradeoff between throughput gain and backhaul load due to the joint transmission of CoMP; C2 indicates that the total number of RBs allocated to UEs associating with a specific BS is limited to the maximum $N_{\mathrm{RB}}$; C3 indicates that aggregate backhaul load of BS *m* should not be over the backhaul capacity $C_{b}^{\max}$.

λ≥0 in Equation (Equation 15) is a coefficient that influences the balance between network throughput and backhaul load. A larger λ suggests that we prefer improving network throughput than reducing backhaul load, and vice versa. Let $\sum_{m\in M^{\Omega }}\log_{2}\left(R_{m}^{\left(0\right)}\right)$ and $\frac{\sum_{m\in M^{\Omega }}V_{m}^{\left(0\right)}}{\left|M^{\Omega }\right|}$ denote sum of logarithm of data rate and averaged load on backhaul links when λ=0, respectively. We define λ with $\sum_{m\in M^{\Omega }}\log_{2}\left(R_{m}^{\left(0\right)}\right)$ and $\frac{\sum_{m\in M^{\Omega }}V_{m}^{\left(0\right)}}{\left|M^{\Omega }\right|}$ as benchmarks, and then λ can be given by:(16)$$\lambda =\frac{\sum_{m\in M^{\Omega }}\log_{2}\left(R_{m}^{\left(0\right)}\right)\left|M^{\Omega }\right|\mu }{\sum_{m\in M^{\Omega }}V_{m}^{\left(0\right)}}.$$
where μ∈[0,1] is a weight factor used for adjusting λ.

The problem in Equation (Equation 15) is a constrained non-convex combinatorial optimization problem, which requires extraordinary high complexity to trace its optimal solution. To obtain a practical solution, we decompose the problem into two steps based on the cross-entropy (CE) method. In this paper, the CE method is chosen because it is a simple, efficient, and general method for solving a great variety of estimation and optimization problems, especially NP-hard combinatorial deterministic and stochastic problems [15]. First, we minimize the backhaul load of the system under the assumption of the conventional *N*-Best user association strategy (By *N*-Best user association strategy, a CoMP UE will associate with $N_{\max}$ BSs which have *N* best SINRs [16]) and propose a content placement algorithm based on cross entropy, which is termed as the CPCE algorithm. Subsequently, under the given content placement strategy, we propose an user association algorithm based on cross entropy, which is referred to as UACE in the rest of this paper.

### 3.2. Cross-Entropy Method

The CE method was originally used in the context of rare event simulation [17] and has been extended as a Monte Carlo method for importance sampling and optimization [17,18].

The principle behind the CE method is to get as close as possible to the optimal importance sampling distribution by using the Kullback-–Leibler (KL) distance as a measure of closeness. By repeatedly updating the Probability Density Function (PDF) of generated samples, the PDF of the samples can finally converge with the obtained optimal strategy solution [19]. Another method with a similar idea is logarithmic loss distortion measure [20,21], where logarithmic loss is also known as cross-entropy loss. The logarithmic loss distortion measure has been recently used in the study of Deep Neural Networks (DNNs) to approach an accurate classifier by minimizing the logarithmic loss. It also performs well on tradeoff between complexity and relevance in representation learning [22].

The main steps of the CE method can be depicted as follows:

**STEP 1: (Encode Strategy Space)**. Consider a UDN constituted by $B^{\Omega }$ BSs, where each BS *b* is a decision-making entity. Suppose that each BS *b* can make a decision out of $N_{b}$ possible strategies, then the strategy set at BS *b* can be expressed as $S_{b}=\left[S_{b}^{1},S_{b}^{2},\cdots ,S_{b}^{N_{b}}\right]$. For a specific decision-making entity BS *b*, $S_{b}^{i}$ is one strategy that belongs to $S_{b}$. The strategy set of $B^{\Omega }$ BSs entities in a UDN can be represented as $S_{B^{\Omega }}=\left[S_{1},S_{2},\cdots ,S_{|B^{\Omega }|}\right]$, which is termed as the strategy space of the CE method. Then the samples of the strategy space of the CE method correspond to the strategies at all BSs in a UDN in current iteration.

Let $P_{b}=\left[P_{b,1},P_{b,2},\cdots ,P_{b,N_{b}}\right]$ denote the probability distribution of sample strategies at BS *b*. Let $P_{b,i}$ denote as the probability of strategy *i* at a specific BS *b*. First, $P_{b,i}$ is initialized to be equal as follows,
(17)$$\begin{array}{c} P_{b,i}=\frac{1}{N_{b}},\;\mathrm{and}\;\sum_{i=1}^{N_{b}}P_{b,i}=1,\left(\forall i=1,\cdots ,N_{b}\right). \end{array}$$

**STEP 2: (Generate Samples According to the Probability).** In the second step of the CE method, sufficient strategy samples should be generated according to the given probability distribution. Denote the *z*th generated sample by $A\left(z\right)=\left[A_{1}^{T}\left(z\right),A_{2}^{T}\left(z\right),\cdots ,A_{b}^{T}\left(z\right),\cdots ,A_{|B^{\Omega }|}^{T}\left(z\right)\right]$, where $A_{b}\left(z\right)$ is the subsample set generated by decision-making entity element *b*. More specifically, the subsample $A_{b}\left(z\right)$ can be represented as:(18)$$\begin{array}{c} A_{b}\left(z\right)=\left[\alpha_{b,1}\left(z\right),\alpha_{b,2}\left(z\right),\cdots ,\alpha_{b,i}\left(z\right),\cdots ,\alpha_{b,N_{b}}\left(z\right)\right], \end{array}$$
where $\alpha_{b,i}\left(z\right)$ is a binary number. For each $A_{b}\left(z\right)$, only one $\alpha_{b,i}\left(z\right)$ is “1” and others are “0” (i.e., $\sum_{i=1}^{N_{b}}\alpha_{b,i}=1$), indicating that only one strategy out of all can be selected at each decision epoch. The probability of subsample $A_{b}$ is $P_{b,i}\in P_{b}$ with $\alpha_{b,i}=1$ and $\alpha_{b,j}=0\left(j\neq i\right)$.

**STEP 3: (Performance Evaluation).** Fitness values of strategy samples can be calculated according to the result of the strategy in the current iteration. Let F(z) denote the fitness value of strategy sample *z*, which can be expressed as:(19)$$\begin{array}{c} F\left(z\right)=-\sum_{b}\sum_{m}x_{mb}\left(1-q_{m}y_{b}^{T}\right)U_{back}. \end{array}$$

Rearrange F(z) in descending order as F(1)≥F(2)⋯F(Z), where *Z* is the maximum number of strategy samples. Then calculate the ρ-quantile of the strategy samples in current iteration $F_{\rho }$ and weed out the unexpected samples. The samples with fitness value $F\left(i\right)\geq F_{\rho }$ are selected for probability updating in the next iteration.

**STEP 4: (Probability Updating).** According to samples selected in the performance evaluation step, the probabilities of each strategy can be updated as follows,
(20)$$\begin{array}{c} P_{b,i}^{\;\mathrm{update}}=\frac{\sum_{z=1}^{Z}I_{F\left(z\right)\geq F_{\rho }}\alpha_{b,i}\left(z\right)}{\sum_{z=1}^{Z}I_{F\left(z\right)\geq F_{\rho }}}, \end{array}$$
where *I* is defined as,
(21)$$I_{x\geq y}=\left\{\begin{array}{cc} 1 & \mathrm{if}\;x\geq y\\ 0 & \mathrm{others} \end{array}\right..$$

Go back to STEP 2, regenerate the samples based on the updated probability distribution and repeat STEP 2 to STEP 4.

**STEP 5: (Convergence Conditions).** The algorithm will come to an end when the fitness value reaches convergence or the algorithm reaches the maximum iteration number set in advance. The cross-entropy method is a global random search procedure, and asymptotical convergence can be achieved to find the optimal solution with probability arbitrarily close to 1 [23,24].

### 3.3. Content Placement Algorithm Based on the Cross-Entropy Method (CPCE)

Before joining in a network, a specific UE will measure Channel State Information (CSI) and choose the candidate BSs with the biggest reference signal received power (RSRP). To investigate the content placement strategies of BSs, we assume the conventional *N*-Best scheme for user association in the first place. BSs in the network are modeled as decision-making entities in the CE method, and feasible content placement candidates are strategies of each entity. The CPCE algorithm can be depicted as Algorithm 1.

The optimized content placement strategy Y can be given by Algorithm 1, under the assumption of the *N*-best user association scheme. User association results can be further optimized with the obtained Y.

**Algorithm 1** Content Placement based on CE method (CPCE)
1:User association under *N*-Best strategy.2:Generate content placement request samples of UEs $q_{m}$ based on Zipf distribution. Map BSs ⟷ Decision-making entities in CE method.3:Map the content placement strategy set ⟷ Strategies in CE method.4:Map the sum backhaul load ⟷ Fitness value in CE method.5:Execute **CPCE**.6:Map the obtained solution into the best content placement strategies of BSs and output Y.


### 3.4. User Association Algorithm Based on the Cross-Entropy Method (UACE)

By Algorithm 1, we obtain the optimal content placement strategy of each BS under the *N*-Best user association scheme. However, the *N*-Best user association scheme does not take into account load balancing and interference management. Under the obtained content placement result, we can further optimize the user association algorithm.

The user association problem is a constrained non-convex integer programming problem, which can also be solved by the CE method [15,19]. Considering that the maximum number of BSs associated with a UE is *N*, the amount of association strategies of a UE will be no more than $2^{N}-1$. Similarly, the steps of the proposed UACE method is as follows.

**STEP 1: (Encode Strategy Space)**. In UACE, the decision-making entities are UEs in the network. Suppose that each UE *m* can make a decision out of $N_{m}$ possible strategies, then the strategy set at UE *m* can be expressed as $S_{m}=\left[S_{m}^{1},S_{m}^{2},\cdots ,S_{m}^{N_{m}}\right]$. Let $P_{m}\;=\left[P_{b,1},P_{b,2},\cdots ,P_{b,N_{m}}\right]$ denote the probability distribution of sample strategies at UE *m*, where $P_{m,i}$ denote the probability of strategy *i* at a specific UE *m*. $P_{m,i}$ can be initialized to be equal as follows,
(22)$$\begin{array}{c} P_{m,i}=\frac{1}{N_{m}},\;\mathrm{and}\;\sum_{i=1}^{N_{m}}P_{m,i}=1,\left(\forall i=1,\cdots ,N_{m}\right). \end{array}$$

**STEP 2: (Generate Samples According to the Probability)**. Samples are generated in this step in a similar way described in STEP 2 of Section 3.2.

**STEP 3: (Encode Strategy Space)**. The fitness value of strategy samples in UACE is
(23)$$\sum_{m\in M^{\Omega }}\log_{10}\left(R_{m}\right)-\lambda \frac{\sum_{m\in M^{\Omega }}V_{m}}{|M^{\Omega }|}$$

**STEP 4 (Probability Updating)** and **STEP 5 (Convergence Conditions)** of UACE are also similar to STEP 4 and STEP 5 in Section 3.2. Then the proposed UACE algorithm can be depicted as in Algorithm 2.

By applying Algorithms 1 and 2, we can obtain suboptimal solutions to problem (Equation 15). With the obtained results, optimized performance of the considered UDN in terms of both throughput and backhaul load is achieved.

**Algorithm 2** User Association based on Cross-Entropy Algorithm (UACE)
1:Execute the proposed CPCE Algorithm 1 under popular contents’ statistics.2:Map UEs ⟷ Decision-making entities in CE method.3:Map association strategy set for a specific UE ⟷ Strategies in CE method.4:Map network throughput of all the cell edge UEs ⟷ Fitness value in CE method.5:Execute **UACE**.6:Map the obtain solution into optimal user association solution and output X.


### 3.5. Complexity Analysis of the Cross-Entropy Method

From the description in the previous subsection, the computational complexity of the proposed CE algorithm is made up of 5 parts.

(1)Initialize the probability distribution of sample strategies. According to the size of encode strategy space and Equation (Equation 18), the computational complexity is $O(|B^{\Omega }|)$;(2)Generate samples according to the probability. According to Equations (Equation 19) and (Equation 20), there are *Z* samples at most, and the size of each sample is $N_{b}\times \left|B^{\Omega }\right|$. Hence, the computational complexity is $O(Z\times N_{b}\times |B^{\Omega }|)$;(3)Performance Evaluation. According to Equation (Equation 21), we should calculate the fitness value of each strategy sample according to Equation (Equation 21), and the computational complexity is $O(Z\times |M^{\Omega }|\times |B^{\Omega }|)$;(4)Probability Updating. According to Equation (Equation 22) and the size of the probability distribution of the sample strategy, the computation complexity is $O(Z\times N_{b}\times |B^{\Omega }|)$;(5)Iteration. The proposed algorithm will come to an end when the maximal iteration number *V* is reached. Hence, the computation complexity is *V* times the sum of the computation complexity from Equations (Equation 1)–(Equation 4), i.e., $O(V\times Z\times N_{b}\times |M^{\Omega }|\times |B^{\Omega }|)$.

According to the analysis above, the total computation complexity of the proposed algorithm based on the CE method is computable in polynomial time.

## 4. Simulation and Analysis

Extensive simulations are conducted to evaluate the performance of the proposed content placement and user association algorithm. In the simulation, 7 MBSs are uniformly distributed in the considered area, while SBSs and UEs randomly drops, the maximum number of BSs that a UE can be associated is 3 [25]. Major parameter settings are listed in Table 1.

### 4.1. System Performance under Different Content Placement Schemes

For content placement strategies, we compared the performance of the proposed CECP scheme to that of random scheme and Most Popular Content (MPC) scheme under different SFs (γ) of popularity of files. Network performance in terms of backhaul load and normalized time delay is shown in Figure 2 and Figure 3 respectively.

When γ=0.2, the backhaul load of the proposed CPCE scheme is two-times less than that of the MPC scheme and the random scheme. As the shape factor increases, the backhaul load and time delay decrease sharply for both the proposed CECP scheme and the MPC scheme. This is because as the shape factor increases, popular files tend to be prone to fewer files, thus more gain can be obtained by selecting proper caching strategies. When γ becomes larger and larger (γ > 0.2 in the simulation), the backhaul load and time delay of the CPCE scheme are comparable to that of the MPC, being about more than 13 times smaller than the Random scheme as shown in Figure 2 and Figure 3.

### 4.2. System Performance of CPCE with Different Numbers of UEs

A simulation was also conducted to evaluate the performance of the proposed CPCE scheme under different network scales with γ=1. As expected, the proposed CPCE algorithm outperforms the MPC and random caching scheme in terms of the backhaul load under different scales of the network, as shown in Figure 4. When UEs are sparsely distributed in the network, the backhaul load of the CPCE scheme is almost ignorable. Even if the number of UEs increases up to 200, the backhaul load of CPCE is still a great deal lower than that of the random scheme and MPC. Figure 5 shows the normalized time delay of each content placement scheme under different network scales. It is observed that CPCE can achieve the lowest time delay compared to the MPC and random caching scheme.

### 4.3. System Performance of CPCE under Different Storage Capacity of BSs

Figure 6 and Figure 7 show the impact on network performance of the BSs’ different storage capacity. It is clear that the more files that BSs can cache, the more possibility that BSs hit required files. We assume the total number of files is 20. When the storage capacity of BSs is half of total files, both backhaul load and time delay of the proposed CPCE is as a third as that of MPC. Even when storage capacity is very limited (for example, only 1 file can be cached), the backhaul load of CPCE is acceptable, as shown in Figure 6. Meanwhile, as shown in Figure 7, time delay decreases as the storage capacity increases, and the proposed CPCE outperforms the other two.

The performance of Random and MPC comparing with the proposed CPCE in terms of time delay and backhaul load is listed in Table 2.

### 4.4. System Performance of CPCE-UACE under Different Weight Factor

The performance of the entire CPCE-UACE algorithm is evaluated in the simulation and compared with *N*-best, No-CoMP, and Threshold user association schemes. No-CoMP scheme means each UE, no matter where it is located, can be associated only to the BS with the best RSRP. Threshold scheme allows a specific UE to be associated with multiple BSs whose RSRP is better than a given threshold [16]. We also assess backhaul load and network throughput of CPCE-UACE algorithm under a different weight factor μ (increases from 0 to 1 by step of 0.5). Generally speaking, the more BSs each UE in the network can be associated with, the better throughput can be achieved, while heavier the backhaul load will be. Fortunately, the proposed CPCE-UACE algorithm can balance backhaul load and network throughput by carefully selecting a weight factor μ. As shown in Figure 8 and Figure 9, network throughput and backhaul load of CPCE-UACE decreases as the weight factor μ grows. When μ is very small (less than 0.05), throughput of CPCE-UACE is significantly better than the others, but the load of it is also outstandingly heavy. On the other hand, when μ is as large as 0.5, both throughput and the backhaul load of CPCE-UACE is lowest in the four schemes considered in the simulation. As a result, we can narrow the range of an optimal μ that perfectly balances network performance in terms of the two aspects into [0.05,0.5]. We consider μ=0.1 as the almost optimal weight factor due to relatively high throughput, as well as the low backhaul load, of CPCE-UACE as shown in Figure 8 and Figure 9.

### 4.5. System Performance of CPCE-UACE under Different Numbers of UEs

In this subsection, we evaluated the proposed CPCE-UACE algorithm jointly in terms of throughput and backhaul load under different network scales with μ=0.1. The number of UEs in the network is set to be from 50 to 200 with an interval 50.

As shown in Figure 10, the average data rate of each UE decreases as the number of UEs in the network grows. It is obvious that the proposed CPCE-UACE algorithm can always achieve an outstanding performance compared to the No-CoMP and N-Best scheme, and 40% performance gain is obtained if $|M^{\Omega }|\leq 150$. Despite the threshold scheme being comparable to the CPCE-UACE algorithm in terms of the average data rate, the threshold scheme has the heaviest backhaul load as shown in Figure 11. It is observed that the proposed CPCE-UACE algorithm has the better performance of the backhaul load compared to the *N*-Best scheme and threshold scheme, as shown in Figure 11. Furthermore, the proposed CPCE-UACE algorithm is comparable with the No-CoMP scheme under a small number of UEs ($|M_{\Omega }|\leq 50$) and outperforms the other schemes in large scale networks ($|M_{\Omega }|\geq 100$).

The performance of No-CoMP, *N*-best, and Threshold compared with the proposed CPCE-UACE in terms of data rate and backhaul load is listed in Table 3.

## 5. Conclusions

This paper considered a problem involving content placement and user association in UDNs where proactive caching and CoMP are enabled. To alleviate the backhaul load and improve network performance, the CPCE-UACE algorithm was proposed to solve the problem. Simulation results demonstrated that the proposed algorithm was capable of decreasing the necessary backhaul traffic and improving network throughput simultaneously. Simulation results showed that the proposed cross-entropy based content placement scheme significantly outperformed the conventional random and MPC placement schemes, with a 50% and 20% backhaul load decrease respectively. Furthermore, the proposed cross-entropy based user association scheme could achieve 30% and 23% throughput gain, compared with the conventional *N*-best, No-CoMP, and Threshold based user association schemes. 

## Figures and Tables

**Figure 1 entropy-21-00576-f001:**
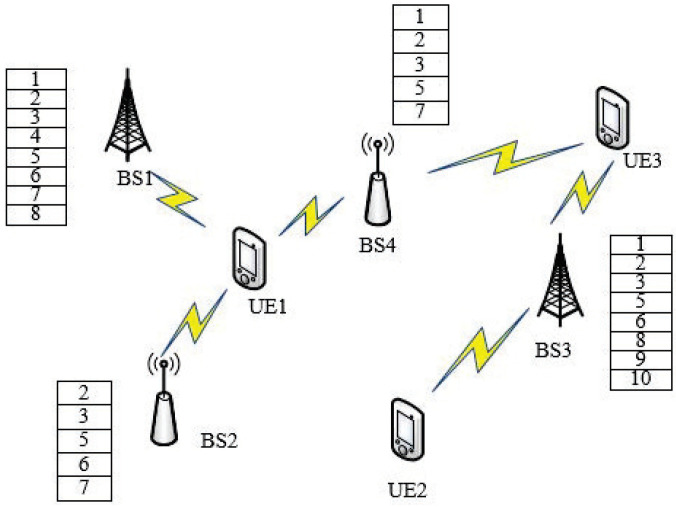
System model of caching-enabled Ultra-Dense Network (UDN) with joint transmission Coordinated Multi-Point (JT CoMP).

**Figure 2 entropy-21-00576-f002:**
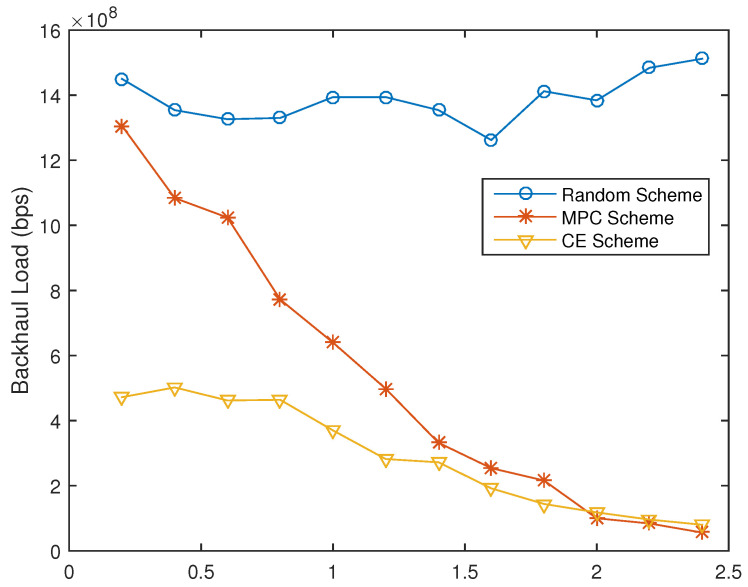
Backhaul load with different γ ($|M^{\Omega }|=200$).

**Figure 3 entropy-21-00576-f003:**
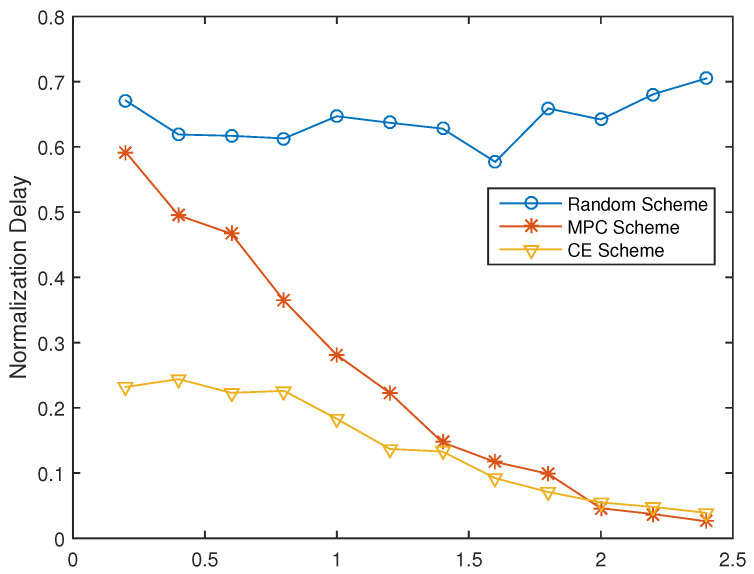
Time delay under different γ ($|M^{\Omega }|=200$).

**Figure 4 entropy-21-00576-f004:**
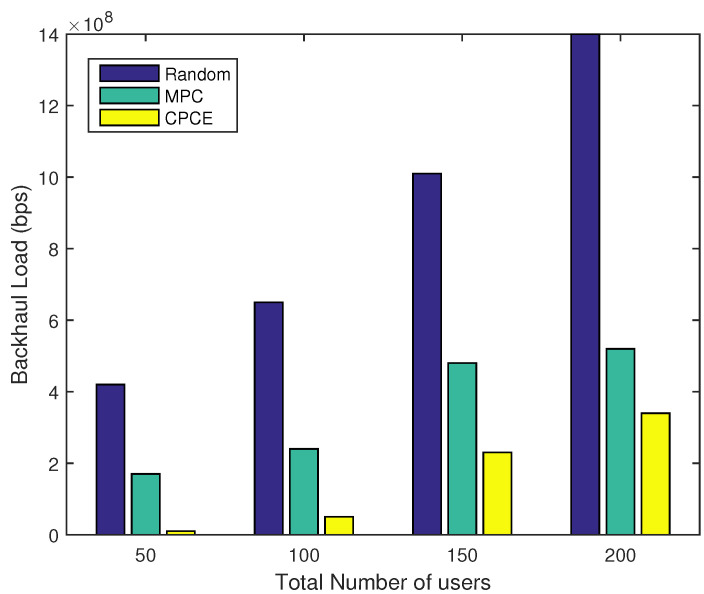
Backhaul load under different numbers of UEs (γ=1).

**Figure 5 entropy-21-00576-f005:**
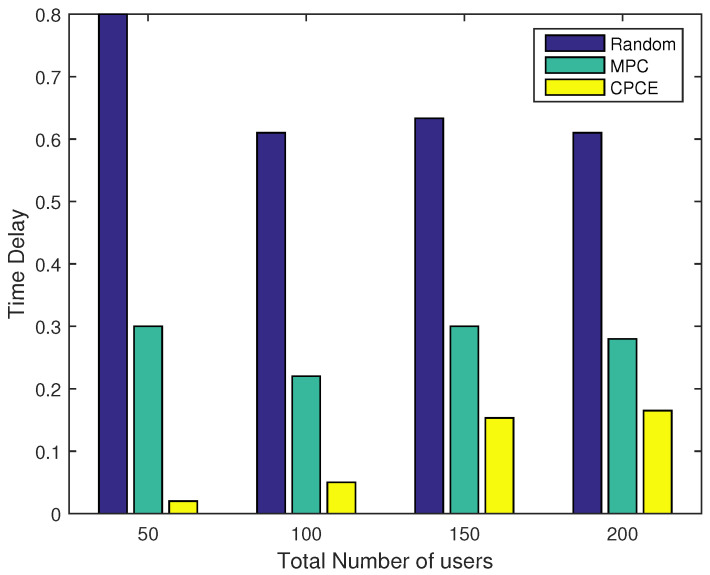
Normalized time delay under different numbers of UEs (γ=1).

**Figure 6 entropy-21-00576-f006:**
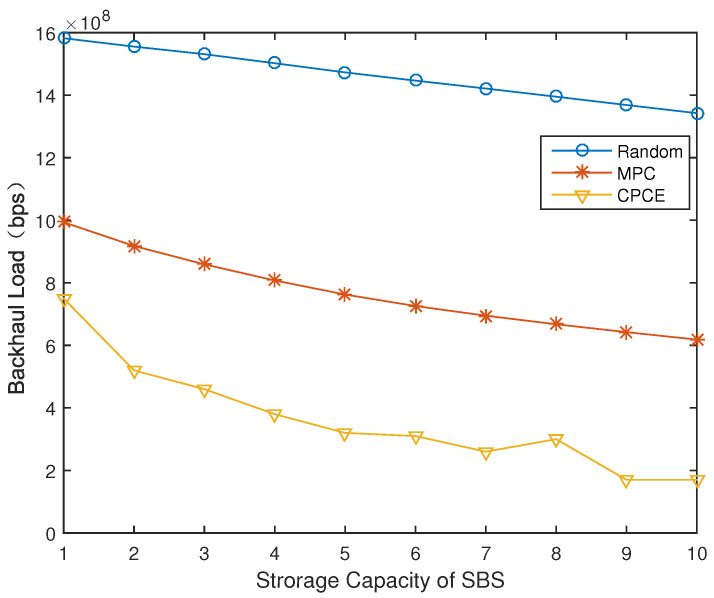
Backhaul load under different storage capacity of BSs (γ=1).

**Figure 7 entropy-21-00576-f007:**
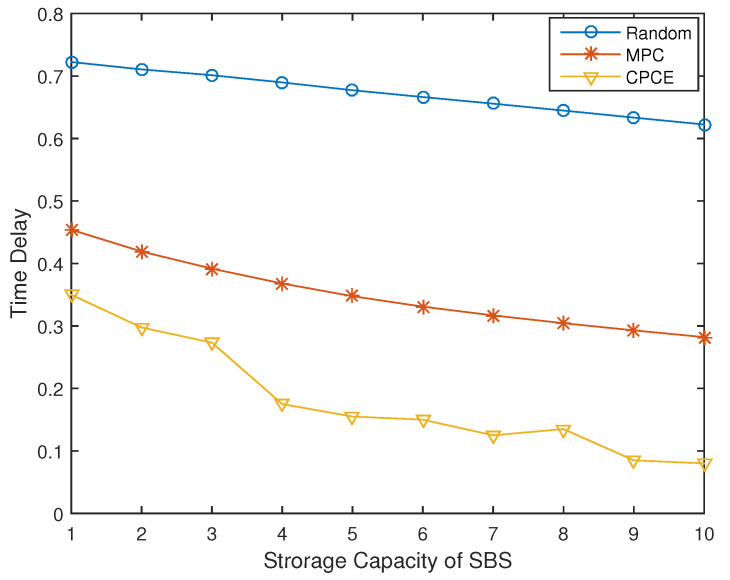
Normalized time delay under different storage capacity of BSs (γ=1).

**Figure 8 entropy-21-00576-f008:**
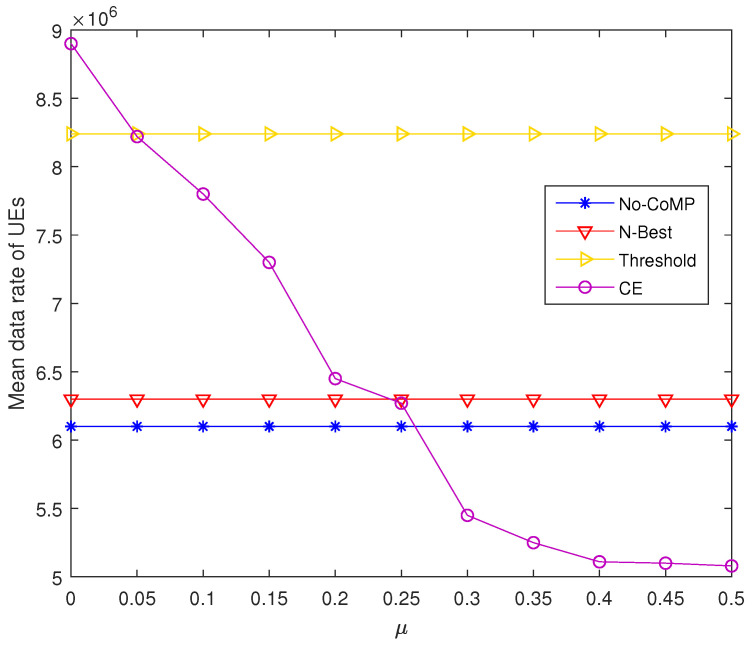
Network throughput under different μ.

**Figure 9 entropy-21-00576-f009:**
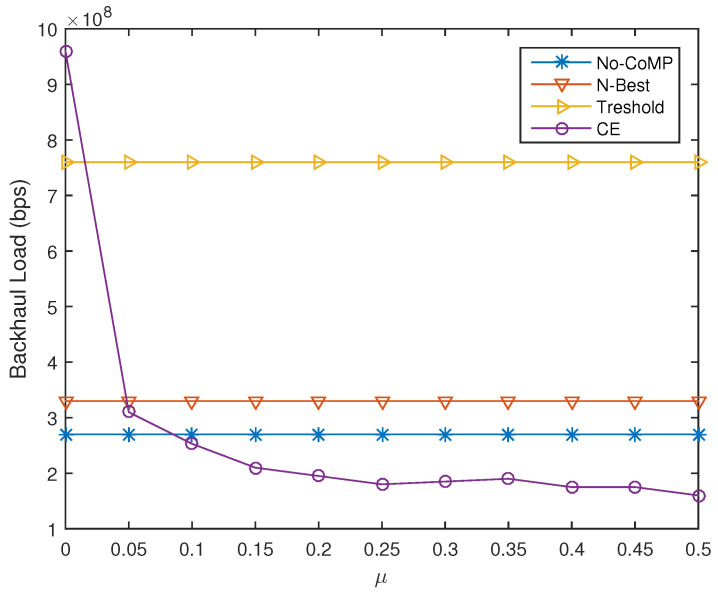
Backhaul load under different μ.

**Figure 10 entropy-21-00576-f010:**
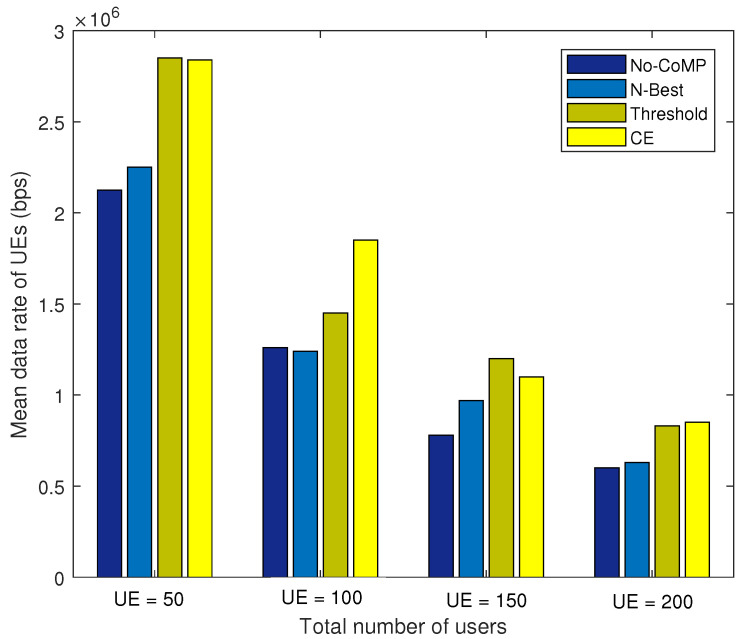
Network throughput under different numbers of UEs.

**Figure 11 entropy-21-00576-f011:**
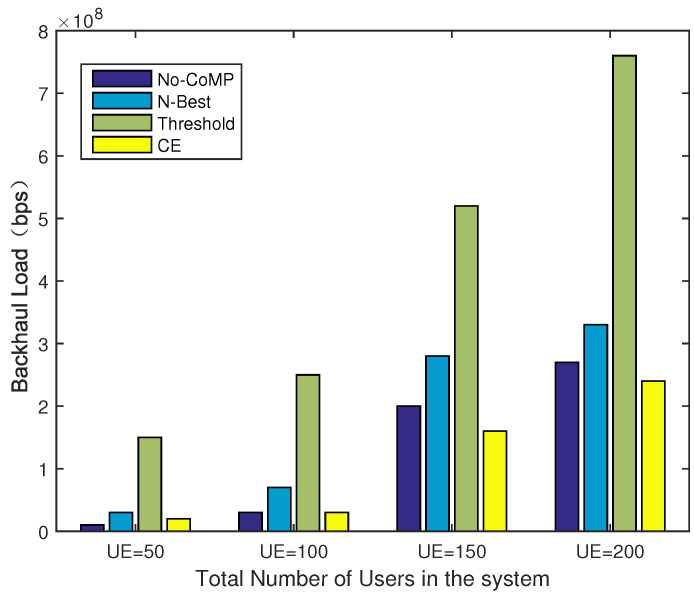
Backhaul load under different numbers of UEs.

**Table 1 entropy-21-00576-t001:** Parameters setting.

Parameters	Value
Plane of Topology	1.5 × 1.5 ${\mathrm{km}}^{2}$
Number of MBSs	7
Number of SBSs	40
Number of UEs	50–200
Channel Model	WINNER
Transmit Power of MBS	40 W
Transmit Power of SBS	2 W
Number of Available RB	100
Total Number of Files	20
Backhaul Capacity of MBS	1 Gbps
Backhaul Capacity of SBS	100 Mbps
Maximal Number of Caching Files on each BS	10
$U_{\mathrm{back}}$	10 Mbps
*N*	3

**Table 2 entropy-21-00576-t002:** Comparison of algorithms in terms of delay and backhaul load.

	Time Delay	Backhaul Load
Random	high	high
MPC	low to high	low to high
CPCE	low	low

**Table 3 entropy-21-00576-t003:** Comparison of algorithms in terms of data rate and backhaul load.

	Data Rate	Backhaul Load
No-CoMP	low	low to medium
*N*-best	low	low to medium
Threshold	medium to high	high
CPCE-UACE	high	low

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
