# Peer review of "Cross-Entropy Method for Content Placement and User Association in Cache-Enabled Coordinated Ultra-Dense Networks"

_entropy, 2019, doi:10.3390/e21060576_

Reviewer 1 Report

The major weakness of this paper is, it's not clear. Many notations are used without explicitly explaining its meaning. This leads to hardness in reading it.  The following is a list of major problems:

$U_m$ is frequently used, but are not introduced at all. Is it same as $R_m$ in (4)? (I am not sure about this). As $U_m$ is not well explained, I am not sure how to obtain (12)?

How to obtain $d_2$ in (13)?  Why take min operation between the two terns in the braces?

In the optimization problem (15), the variables are X, Y. But the condition C3, it does not involve any variable about X and Y.  With my understanding, $q_{m,f}$ should be randomly generated parameters according to some distribution, such as  (6). So it is NOT the parameter that can be optimized.

In (16), the notations $M_E$ and $V_{total}$ are used without any explanation. 

I am lost from Section 3.2. What is the connection between the proposed problem formulation and the Cross Entropy Method? What is a "strategy" mentioned in STEP 1? The notation $N_b$ is again a notation used but not explained .

The notation $y_b$ should be the b-th row vector in (8) I guess. But it should be explained when used.

In (15), the condition C1, is the notation $N_max$ the same with $N$ in Page 3, Line 99? 

Why used two different notations here?

The authors mentioned Pico BSs at the begining of Section 2.1. What is Pico BS? Is it same as SBS? Is it one type of BS aside from Mocro BS and Small BS? Is it contained in $B^{\Omega}$?

Typos: Page 3, last paragraph:"Each UE in the network can occupied..."->"Each UE in the network can occupy..."

Reviewer 3 Report

Authors in this paper discuss the coordination of data distribution in an ultra-dense network to improve the network performance in terms of throughput and latency. The problem definition given is clear and authors made an attempt to consider all the factors involved.

Introduction, and more specifically the third paragraph needs to be explained better and clearer. Although authors collected lot of contents for the introduction, it does not have a coherent structure.

Comparison to other works in the literature can be explained better in the results.

It would be helpful if the authors can characterize the main contributors and effect of each on the data rate and latency in the conclusion.  

The computational complexity of the cross-entropy method is not discussed. This is particularly important if the algorithm is supposed to run in real-time applications

Author Response

Round  2

Reviewer 1 Report

The paper proposes use Cross-Engropy Method to solve the problem of Content Placement and User Association problem in Ultra Dense Networks (UNDs).  

Firstly, I think the model is not clear enough:

Why  introduce both "Macro BSs" and "Small BSs". From the technical part of this paper, they have no distinction. To simplify the model, I suggest just mention "BSs".

In (4), why ceilling $\beta_m N_{RB}$. From achievability view, I think it should be floor. 

In Line 128, the authors mentioned "each file is in the same size of S bits". But in Line 158, the authors introduced another symbol for file size $F_{\max}$. Why introduce redundant symbols?

I can't under stand how to get to  (13). What is the meaning of $F_{\max}/U_{back}$, why it can be used to measure delay?

In the optimization formulation (15), I can't understand why take logorithm on the rate $R_m$? Notice in (4), $R_m$ is the rate, which is measured in bits.

Why $\lambda$ can be given in (16)? Plugging (16) into (15), (15) will become $(1-\mu)\sum log2(R_m)$, there is no tradeoff between rate and backhaul load. 

In addition, the algorithms are also not clear enough:

The authors describes a Cross Entropy Method framework in Section 3.2, and then cite it in Algorithm 1 and Algorithm 2. The questions are:

    1. In the Cross Entropy Method description, the notion "Elements" is not mentioned at all. So, I can't under stand "Elements" in both Algorithm 1 and Algorithm 2. 

     2. It seems that, the Cross Entropy Method depicted in Section 3.2  is tailored for Algorithm 1, becuse in Step 1, it mentions notions such as BSs. But how to adapt it to Algorithm 2? Things are completely different there. 

     With above reasons, I  feel that it is not a well prepared manuscript. 
